# An Investigation on CCT and Ra Optimization for Trichromatic White LEDs Using a Dual-Weight-Coefficient-Based Algorithm

**DOI:** 10.3390/mi13020276

**Published:** 2022-02-09

**Authors:** Hua Xiao, Yan Li, Binghui Li, Guancheng Wang

**Affiliations:** 1School of Electronic and Information Engineering, Guangdong Ocean University, Zhanjiang 524088, China; wanggc@gdou.edu.cn; 2Technology Development Centre, Shenzhen Institute of Guangdong Ocean University, Shenzhen 518120, China; 3Research and Development Center for Solid State Lighting, Institute of Semiconductors, Chinese Academy of Sciences, Beijing 100083, China; yanli7@semi.ac.cn; 4Shenzhen Institute of Artificial Intelligence and Robotics for Society (AIRS), The Chinese University of Hong Kong (CUHK), Shenzhen 518000, China; libinghui@cuhk.edu.cn

**Keywords:** light-emitting diode, spectral optimization, correlated color temperature (CCT), general color rendering index (Ra)

## Abstract

Spectral optimization is applied as an effective tool in designing solid-state lighting devices. Optimization speed, however, has been seldomly discussed in previous reports as regards designing an algorithm for white light-emitting diodes (WLEDs). In this study, we propose a method for trichromatic WLEDs to obtain the optimal Ra under target correlated color temperatures (CCTs). Blue-, yellow-, and red-color monochromatic spectra, produced by the GaN LED chip, YAG:Ce^3+^ phosphors, and CdSe/ZnSe quantum dots, respectively, are adopted to synthesize white light. To improve the effectiveness of our method, the concept of dual weight coefficients is proposed, to maintain a numerical gap between the proposed floating CCT and the target CCT. This gap can effectively guarantee that Ra and CCT ultimately move toward the targeting value simultaneously. Mechanisms of interaction between CCT, Ra, and dual-weight coefficients are investigated and discussed in detail. Particularly, a fitting curve is drawn to reveal the linear relationship between weight coefficients and target CCTs. This finding effectively maintains the accuracy and accelerates the optimization process in comparison with other methods with global searching ability. As an example, we only use 29 iterations to achieve the highest Ra of 96.1 under the target CCT of 4000 K. It is hoped that this study facilitates technology development in illumination-related areas such as residential intelligent lighting and smart planting LED systems.

## 1. Introduction

In comparison with RGB LEDs, light-conversion-material-based white light-emitting diodes (WLEDs) have played leading roles in solid-state illumination, due to their high light-conversion efficiencies in specific wavebands, stability under various junction temperatures, low cost, and feasibility in color tunability [1,2,3]. Conventionally, the color performance of WLEDs is evaluated from two aspects: the emitting color from the WLED and the releasing color of objects exposed under the WLED. Theoretically, the former is characterized by correlated color temperature (CCT), and the latter is characterized by color rendering property, in which the concept of general color rendering index (Ra) is conventionally adopted for evaluation by the International Commission on Illumination [4]. CCT expresses a warm or cold feeling when we observe the light beam, while Ra reveals the ability of a light source to express the real color of an object. Ra represents the average value of the color rendering index (CRI) of eight general colors in a WLED system. CCT and Ra are functions of monochromatic spectral power distributions (SPDs) of different colors [5]. Therefore, if we intend to adjust the circadian rhythm of humans and plantings or reduce driver’s fatigue in a specific scenario, modulating the SPD of the WLED system is effective and indispensable [6]. In trichromatic WLEDs, the white-light SPD (SPD*_W_*(λ)) is conventionally generated by downconverting light-conversion materials with blue LED chips, in which light in short wavebands (such as blue light) can be effectively converted to light in long wavebands (such as red light) under the stokes effect.

In the last few decades, SPD optimization technologies have been widely investigated to facilitate illumination in areas of residential lighting [7], agriculture [8], rehabilitation therapy [9], and visible light communication [10,11]. These technologies can be mainly divided into two catalogs: the first type is to optimize the color rendering property and energy consumption by adjusting peak wavelengths, spectral bandwidth, and intensity by using Gaussian functions; the other is to optimize the color rendering property and energy consumption by adjusting the density of real light in different colors. For the first type, Guo et al. [12] conducted comprehensive numerical simulations of three-hump and four-hump SPDs in WLEDs. The changes in Ra and CCT values with the shifting peak wavelengths of full width at half maximums (FWHMs) were analyzed under different operating temperatures. The relationship between scotopic–photopic ratio and CCT was investigated as well. Wei et al. [13] proposed six-channel-based LEDs to synthesize daylight with high quality by using a genetic algorithm and Gaussian spectral model. For the second type, Zhu et al. [14] conducted a comprehensive study on illumination performances of the perovskite-based LED with four humps. Titkov et al. [15] proposed a semi-hybrid device, which combined monolithic blue-cyan LED with green-red phosphor mixture, exhibiting the highest Ra of 98.6 at CCT of 3400 K. Yuan et al. [16] manufactured a trichromatic WLED, which constitutes of blue-pump carbon dots and phosphor glass, realizing the highest Ra of 92.9 at CCT of 3610 K. Among these studies, a variety of methods are conventionally used, such as the multiple Gaussian function method [17], least-squares method [18], and iterative method of gradient descent [19]. However, these methods focus on improving the accuracy and the feasibility, as well as developing light-conversion material species with superior chromaticity; few of them discuss the improvement strategy of optimization speed for WLEDs.

With the development of the Internet of Things (IoT) and 5G technologies, the intelligent control technology of illumination lamps becomes imperative for saving energy and increasing productivity [20,21]. Therefore, improving the effectiveness of spectral optimization becomes a key issue in intelligent control. In this study, we propose a convenient method to optimize CCT and Ra values simultaneously for trichromatic WLEDs by using dual-weight coefficients. These coefficients can effectively control the variation range of CCT while searching for the optimal Ra value. Key steps to realize the proposed method is analyzed comprehensively. Compared with other conventional methods used for spectral optimization, the proposed method can greatly accelerate the calculation process while maintaining accuracy.

## 2. Monochromatic Spectra Preparation and Theory of Algorithms

As shown in Figure 1a, the blue-emissive LED chip (302 × 198 µm^2^, Hualian Co., Ltd., Xiamen, China) was selected as the excitation source, and yellow-emissive cerium-doped yttrium aluminum garment phosphors (YAG:Ce^3+^, Youyan Rare Earth Co., Ltd., Beijing, China) and red-emissive CdSe/ZnSe quantum dots (Poly OptoElectronics Co., Ltd., Jiangmen, China) were selected as light-conversion materials to fabricate white light. YAG:Ce^3+^ phosphors can greatly broaden the white-light spectrum in the visible-light regime, while CdSe/ZnSe quantum dots are able to provide pure red emission with high stability and high quantum yields (QYs). Recently, QYs of YAG:Ce^3+^ phosphors and CdSe/ZnSe quantum dots can reach up to 90% and ~100%, respectively [22,23]. Packaging technology of the WLED has been given in [24]. To facilitate our study, spectra of monochromatic blue, yellow, and red light are referred to as SPD*_B_*(λ), SPD*_Y_*(λ), and SPD*_R_*(λ), respectively.

A 500 mm diameter-integrating sphere (Everfine) was utilized to measure SPD*_B_*(λ). To obtain SPD*_Y_*(λ) and SPD*_R_*(λ), we mathematically removed the superposition area of blue light of the original emission spectra produced by phosphors and quantum dots [24]. SPD*_B_*(λ), SPD*_Y_*(λ), and SPD*_R_*(λ) were normalized before optimization. From Figure 1b, we observe that the LED chip and CdSe/ZnSe quantum dots generate narrow blue and red peaks with FWHM of 54 nm and 56 nm, respectively. On the other hand, YAG:Ce^3+^ phosphors produce a spectrum with FWHM of 125 nm that covers a wide range of visible light. Here, we assumed that the peak wavelength and the FWHM of these monochromatic spectra are independent of the driven current, so SPD*_W_*(λ) can be described as a linear combination of SPD*_B_*(λ), SPD*_Y_*(λ), and SPD*_R_*(λ), as described by
(1)$${\mathrm{SPD}}_{W}\left(\lambda \right)=A_{B}\cdot {\mathrm{SPD}}_{B}\left(\lambda \right)+A_{Y}\cdot {\mathrm{SPD}}_{Y}\left(\lambda \right)+A_{R}\cdot {\mathrm{SPD}}_{R}\left(\lambda \right)$$
where $A_{B}$, $A_{Y}$, and $A_{R}$ are the proportions of the radiant power of blue, yellow, and red light, respectively.

Before calculation, target CCT, test CCT, and test Ra values are defined as CCT_tar_, CCT_test_, Ra_test_, respectively. CCT_test_ and Ra_test_ represent current CCT and Ra values in the calculation. Figure 2a illustrates the steps for spectral optimization using the proposed method. For comparison, conventional methods I and II used for spectral optimization are illustrated in Figure 2b,c. Among these three methods, method I directly considers all the possibilities of $A_{B}$, $A_{Y}$, and $A_{R}$ under CCT_tar_, while method II randomly selects values of $A_{B}$, $A_{Y}$, and $A_{R}$ until fulfilling the cycle index, which is set as 1000 for methods I and II. Both methods I and II use the bubbling method to obtain the highest Ra within the error range of CCT_tar_. For the proposed method, the calculation steps are described as follows:

(1)First, we initialize the procedure and load original data, such as the spectra of monochromatic light, step lengths for iteration, error ranges of Ra and CCT, and initial values of $A_{B}$, $A_{Y}$, and $A_{R}$;(2)Two key problems for CCT optimization are how to adjust CCT_test_ and how to optimize Ra in the meantime. According to the relationship between CCT and components of different colors, we first set a floating parameter between the initial CCT and CCT_tar_, which is named CCT_m_. The relationship between CCT_m_ and CCT_test_ can be expressed as CCT_test_ = δ_1_ + CCT_m_, where δ_1_ is the first weight coefficient in our algorithm. To realize CCT_m_, we only need to modulate the parameter of $A_{B}$;(3)Before realizing CCT_tar_, we optimize Ra_test_ by using the bubbling method. Keeping the proportion of $A_{B}$ and $A_{Y}$ unchanged, we attempt to modulate $A_{R}$ with a small step to observe the change of Ra_test_. If the small step helps to increase the value of Ra, we conduct a similar iteration until Ra_test_ reaches the highest value; otherwise, we modulate $A_{R}$ in the negative direction. The relationship between CCT_m_ and CCT_tar_ can be expressed as CCT_tar_ = δ_2_ + CCT_m_, where δ_2_ is the second weigh coefficient in our algorithm.(4)When the calculation result meets the required conditions, we export optimized $A_{B}$, $A_{Y}$, and $A_{R}$ values, the optimized WLED spectra, CCT_test_, as well as Ra_test_.

Below is the design philosophy of the proposed algorithm. Particularly, we propose a floating CCT value named CCT_m_ and two weight coefficients, named δ_1_ and δ_2_, to control the variation range of the CCT_test_. There exist two main stages in the optimization process: We first impel CCT_test_ to move toward CCT_m_ and then optimize Ra_test_ and CCT_test_ simultaneously, to reach the optimum Ra and CCT_tar_. If we directly search for CCT_tar_, the variation space of Ra_test_ is very limited, due to the interaction effect of CCT and Ra. The proposal of CCT_m_ can effectively solve this problem, rendering CCT_test_ reach a position near CCT_tar_ before the optimization of Ra_test_.

Weight coefficients of δ_1_ and δ_2_, which determine the value of CCT_m_ and the shifting range of CCT_test_, are key for the optimization result. If δ_2_ is too small, Ra_test_ will not reach the highest value due to the limited shifting space; on the other hand, if δ_2_ is too large, Ra_test_ can reach the highest value soon but at the expense of the error between CCT_test_ and CCT_tar_. Another problem is how to guarantee that CCT_test_ moves toward CCT_tar_ instead of the reverse direction while optimizing Ra_test_. To solve this problem, we set the original $A_{B}$, $A_{Y}$, and $A_{R}$ values as 0.1, 0.3, and 0.5, respectively, in which $A_{R}$ is large enough to guarantee the decreasing trend of $A_{R}$ while optimizing Ra_test_.

## 3. Results and Discussion

### 3.1. Relationship between CCT, Ra, and Other Parameters

Figure 3 illustrates the variation of CCT_test_, Ra_test_, and δ_1_ under different δ_2_ in a 3D coordinate diagram, when CCT_tar_ is set as 8000 K. To facilitate discussion, these results are separately presented in Figure 3a,b at different view angles. In Figure 3a, CCT_test_ is decreasing with the increase in δ_1_ when δ_2_ equals 3000 K. The reason is that δ_1_ directly influences the difference between CCT_test_ and CCT_m_. If δ_1_ is too large, CCT_test_ becomes much smaller than CCT_m_, increasing the difficulty of reaching CCT_tar_ while optimizing Ra_test_. According to the methodology of the algorithm, optimization will finally stop when we obtain the optimal Ra. By then, the final CCT_test_ obtained may fail to reach CCT_tar_.

Secondly, Ra_test_ increases with the increase in δ_2_. Since δ_2_ represents the difference between CCT_m_ and CCT_tar_, when we increase δ_2_, CCT_m_ becomes smaller. Thus, larger δ_2_ provides a wider range for Ra optimization, extending the shifting area of Ra_test_ within the permitted range of CCT_test_.

From Figure 3b, we observe the variation of CCT_test_, Ra_test_, and δ_1_ under different δ_2_ at the other view angle. When δ_2_ is set as 600 K, 1200 K, and 1800 K, respectively, the curves almost lie in a similar plane with CCT_tar_, equal to 8000 K. However, for δ_2_ = 2400 K and δ_2_ = 3000 K, corresponding curves stretch out of this plane. In other words, their CCT_test_ become much smaller than CCT_tar_ of 8000 K. Below are the explanation for this phenomenon. For δ_2_ = 2400 K and δ_2_ = 3000 K, we reserve a large variation range of CCT to support the optimization for CCT_m_ and Ra_test_, causing the inaccessibility of CCT_tar_ when the optimization process of Ra is ending. This explains the phenomenon that those points on the curve are almost far away from 8000 K when δ_2_ = 3000 K. When δ_2_ decreases from 3000 K to 2400 K, a portion of points on the curve return to the plane of 8000 K. To summarize, those points staying near the plane of 8000 K are constrained by CCT_tar_; those points that stretch out from the plane of 8000 K are constrained by optimization conditions of Ra.

As shown in Figure 4a, R_atest_ increases with an increase in δ_1_ and δ_2_. If δ_1_ remains unchanged and δ_2_ decreases, it would cost more iteration steps to increase $A_{B}$ in order to improve CCT_test_. Hence, even CCT_test_ reaches CCT_m_; however, the value of $A_{B}$ already becomes very large, which limits the highest Ra the WLED can realize. As lower $A_{B}$ helps the prompt enhancement of Ra_test_, overlarge $A_{B}$ hinders the improvement of Ra_test_, despite the adjustment of $A_{R}$. On the other hand, if we keep δ_2_ unchanged and decrease δ_1_, CCT_test_ will reach CCT_m_ sooner; however, Ra_test_ cannot be fully optimized. Therefore, the value of optimal $A_{B}$ is influenced by CCT_m_ and is finally determined by δ_1_ and δ_2_. Increasing δ_1_ and δ_2_ under high CCT_tar_ can greatly decrease CCT_m_, enlarging the optimization range of Ra. In Figure 4b, CCT_test_ slightly decreases when we increase δ_1_ or δ_2_, indicating that CCT_test_ has a reverse shifting trend, compared with Ra_test_ under different δ_1_ and δ_2_ values.

These analyses reveal the significance of δ_1_ and δ_2_ for the optimization result. The WLED with different CCT_tar_ values has different reactions under similar δ_1_ and δ_2_. Thus, balancing the relationship between δ_1_, δ_2_, and CCT_tar_ is the next step to accelerate the optimization process.

Figure 5a illustrates the optimized spectra of the trichromatic WLED under conditions of CCT_tar_ = 8000 K and δ_1_ = 200 K. When δ_2_ increases from 600 K to 3000 K, peaks of blue and red light slightly decrease, while Ra_test_ increases from 66.9 to 89.7. This is because the WLED spectrum is increasingly close to the spectrum of the reference source (black body source) [14]. The shifting trend of Ra_test_ matches well with the analysis results of Figure 3a.

Figure 5b describes the shifting trend of CCT_test_ and Ra_test_ in the iteration process under various CCT_tar_ values. When CCT_tar_ ranges from 4000 K to 8000 K, Ra_test_ declines slowly at first, as shown in step 1; thereafter, Ra_test_ abruptly decreases in a small step, as shown in step 2; finally, Ra_test_ increases severely until reaching the top point before finishing the optimization steps, as shown in step 3. With the increase in CCT_tar_, the highest value of Ra that can be achieved decreases. A similar phenomenon has been observed in [24,25].

When CCT_tar_ ranges from 4000 K to 8000 K, CCT_test_ also shows three steps to reach CCT_tar_; however, the shifting trend of CCT_test_ during the optimization process is different from that of Ra_test_. Ra_test_ decreases slowly at first and then increases drastically; on the other hand, CCT_test_ increases slowly at first and then increases drastically. A comparison of Figure 5a with Figure 5b indicates that the increase in CCT_test_ in step1 sacrifices the improvement of Ra_test_ in the initial time. In step 3, different from the optimization aim of Ra_test_, we only need to find a CCT value near CCT_tar_ instead of finding the local optimal value of CCT_test_. It is worth noting that only three iterations are used for optimization when CCT_tar_ equals 3000 K, and the value of iterations increases with the increase in CCT_tar_. This is because the initial values of $A_{B}$, $A_{Y}$, and $A_{R}$ are very close to optimized values of $A_{B}$, $A_{Y}$, and $A_{R}$ under low CCT_tar_.

As we mentioned in Figure 3, different optimization results can be obtained under different CCT_tar_ with similar δ_1_ and δ_2_ values. To guarantee the achievement of the optimal Ra_test_ in all cases, we should initially manage to acquire optimized values for δ_1_ and δ_2_ (δ_Opt1_ and δ_Op2_) under different CCT_tar_ values. It is worth noting that there exists a strong relationship between the sum of δ_Opt1_ and δ_Opt2_ (∑(δ_Opt1_, δ_Opt2_)) and CCT_tar_. As shown in Figure 6, ∑(δ_Opt1_, δ_Opt2_) is plotted and fitted using a linear function under different CCT_tar_ values, which ranges from 3000 K to 12,000 K. It is evident that ∑(δ_Opt1_, δ_Opt2_) presents a perfect linear increasing trend with the increase in CCT_tar_. The slope of this curve is calculated by using the linear interpolation method, and the curve can be described as
(2)$$\sum \left(\delta_{\mathrm{Opt}1},\delta_{\mathrm{Opt}2}\right)=\;\alpha \cdot {\mathrm{CCT}}_{\mathrm{test}}$$
where α is calculated to be 0.63 for the proposed WLED. For WLEDs combined with different light-conversion materials, the numerical value of α should be different. Additionally, the measured data slightly deviates from the fitting curve when CCT_tar_ equals 3000 K, which is probably because changes in δ_1_ and δ_2_ do not have visible effects on the optimization result when CCT_tar_ is low.

Once the law between α and light-conversion materials is identified, it is necessary to select the optimal δ_1_ and δ_2_ under different CCT_tar_ values before optimization. According to Figure 4, the optimal Ra_test_ corresponds to the largest δ_1_ and δ_2_ within the allowed range of CCT_tar_. Except for the linear relationship between ∑(δ_Opt1_, δ_Opt2_) and CCT_tar_, the value of δ_2_ should be larger than δ_1_, to guarantee the operation of the calculation procedure. Therefore, we had better select larger δ_2_ and smaller δ_1_ to satisfy Equation (2). This principle provides us with an effective way to accelerate the spectral optimization speed.

### 3.2. Comparison between the Proposed Method, Method I, and Method II

Table 1, Table 2 and Table 3 present calculation parameters of spectral optimization with the proposed method, method I, and method II. In Table 1, optimized CCT_test_ values are very close to CCT_tar_. Among all results, the highest Ra_test_ reaches up to 96.1, with CCT_test_ of 4013 K. $A_{B}$, $A_{Y}$, and $A_{R}$ exhibit a regular shifting trend in which $A_{B}$ increases, and $A_{R}$ reduces with the increase in CCT_tar_. The sum of δ_1_ and δ_2_ increases with the increasing CCT_tar_, which is consistent with the discussion and results in Figure 6. Optimization results of the proposed method and method I, in terms of CCT_test_, Ra_test_, $A_{B}$, $A_{Y}$, and $A_{R}$ values, are highly coincident with each other. This coincidence verifies the correctness of the proposed method.

In Table 3, calculation results of method II under low CCT_tar_ well match those of method I. Compared with the proposed method and method I, we can even obtain better optimization results of Ra_test_ under 3000 K by using method II. However, with the increase in CCT_tar_, method II fails to effectively improve Ra_test_ to obtain the optimal value. Due to the random selection rule of method II, calculation results of $A_{B}$, $A_{Y}$, and $A_{R}$ listed in Table 3 do not show a similar trend as in Table 1 and Table 2. These results reveal that method II cannot effectively optimize WLED spectra under high CCTs. To apply $A_{B}$, $A_{Y}$, and $A_{R}$ in a real scenario for realizing target illumination effects, ref. [24] presented the implementation method in detail.

By using Equation (2) to find δ_Opt1_ and δ_Opt2_ under different values of CCT_tar_, we accelerate the optimization process. In Figure 7a, the number of iterations of these three methods under different values of CCT_tar_ is compared. Obviously, the number of iterations of the proposed method is much less than that of the other two methods. For the proposed method, the number of iterations increases with the increase in CCT_tar_. The accuracy of CCT_test_ and Ra_test_ for these three methods can be evaluated by using the error range concept. Error ranges of CCT_test_ and Ra_test_ (ε_C_ and ε_R_) are calculated by |CCT_tar_ − CCT_test_|/CCT_tar_ and |100 − Ra_test_|/100, respectively. ε_C_ and ε_R_ under different values of CCT_tar_ are given in Figure 7b,c for comparison. The ε_C_ values of these three methods are comparable under different values of CCT_tar_. The ε_R_ values of the proposed method and method I are similar, and they are relatively smaller than that of method II under high CCT_tar_ values. These results verify the effectiveness and accuracy of the proposed method.

## 4. Conclusions

In this study, we propose an effective method to optimize the Ra of trichromatic WLEDs under different CCTs. Compared with conventional methods I and II, the proposed method exhibits superior searching ability to find the optimal Ra under target CCTs. Specifically, the highest Ra of 96.1 under 4013 K can be obtained after only 29 iterations. Three main mechanisms were investigated and analyzed for the proposed method: (1) the influence of δ_1_ and δ_2_ on the calculation results of CCT_m_, CCT_test_, and Ra_test_; (2) the relationship between δ_1_, δ_2_, CCT_m_, $A_{B}$, $A_{Y}$, and $A_{R}$; (3) the shifting rule of δ_Opt1_ and δ_Opt2_ under different CCT_tar_ values. Particularly, the fitting linear curve that describes the relationship between ∑(δ_Opt1_, δ_Opt2_) and CCT_tar_ can provide an effective way to greatly accelerate the optimization process under different CCT_tar_ values. This study reveals the shifting mechanism of CCT and Ra values with dual-weight coefficients and greatly enhances the effectiveness of spectral optimization for WLEDs. Our method is hopefully applied in related areas such as residential intelligent lighting and smart planting LED systems.

## Figures and Tables

**Figure 1 micromachines-13-00276-f001:**
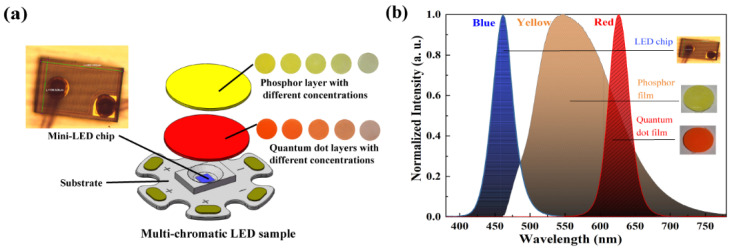
(**a**) The assembly of the trichromatic LED sample and (**b**) monochromatic spectra of blue, yellow, and red light, which produced by LED chip, YAG:Ce^3+^ phosphors, and Cdse/ZnSe quantum dots, respectively.

**Figure 2 micromachines-13-00276-f002:**
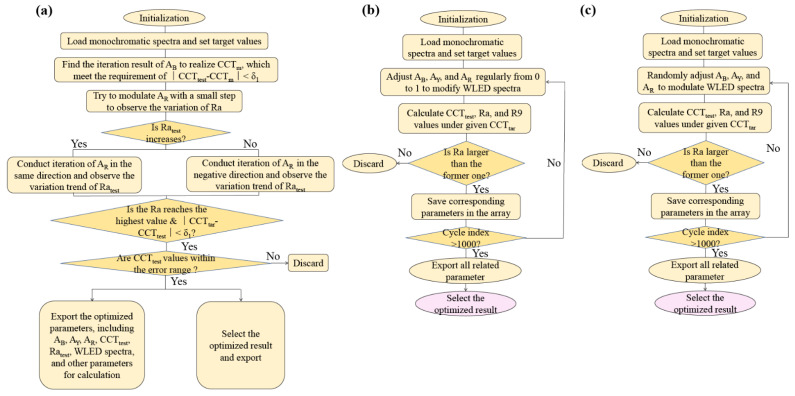
Flow diagrams of (**a**) the proposed method, (**b**) method I, and (**c**) method II, in which methods I and II are conventionally used for spectral optimization.

**Figure 3 micromachines-13-00276-f003:**
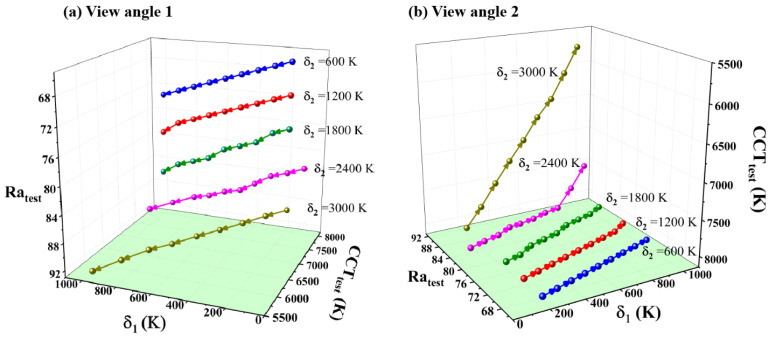
Shifting trend of CCT_test_, Ra_test_, and δ_1_ under different δ_2_ in a 3D coordinate diagram with CCT_tar_ of 8000 K. (**a**,**b**) are the same 3D figure at different view angles.

**Figure 4 micromachines-13-00276-f004:**
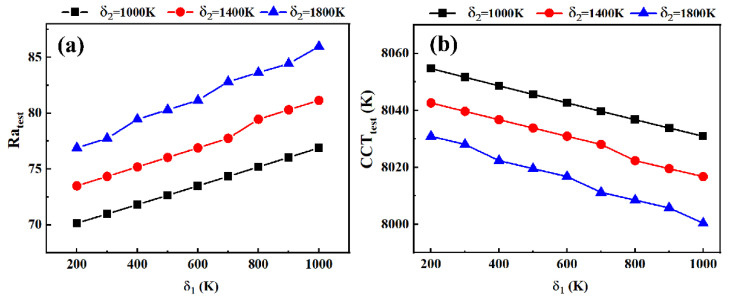
(**a**) Variation in Ra_test_ under different δ_1_ values, when δ_2_ equals to 1000 K, 1400 K, and 1800 K, respectively; (**b**) variation in CCT_test_ under different δ_1_ values, when δ_2_ equals to 1000 K, 1400 K, and 1800 K, respectively.

**Figure 5 micromachines-13-00276-f005:**
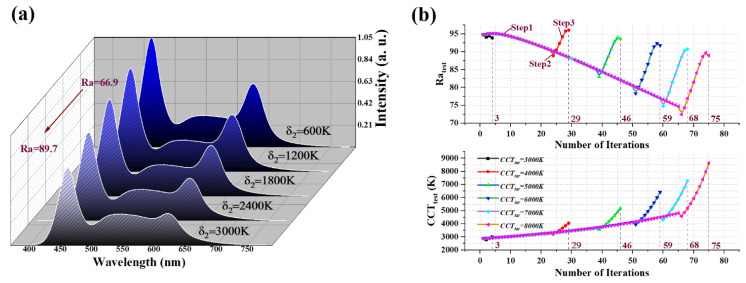
(**a**) The optimized spectra of the trichromatic WLED under the condition of CCT_tar_ of 8000 K and δ_1_ of 200 K. (δ_2_ is ranging from 600–3000 K); (**b**) the shifting trend of iterations of CCT_test_ and Ra_test_ under different CCT_tar_ values.

**Figure 6 micromachines-13-00276-f006:**
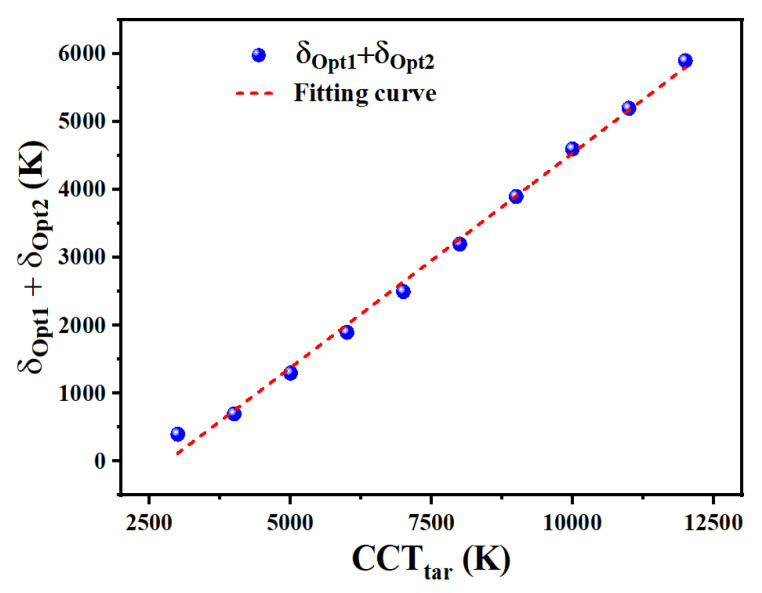
Data and fitting curve of ∑(δ_Opt1_, δ_Opt2_) under different CCT_tar_ values.

**Figure 7 micromachines-13-00276-f007:**
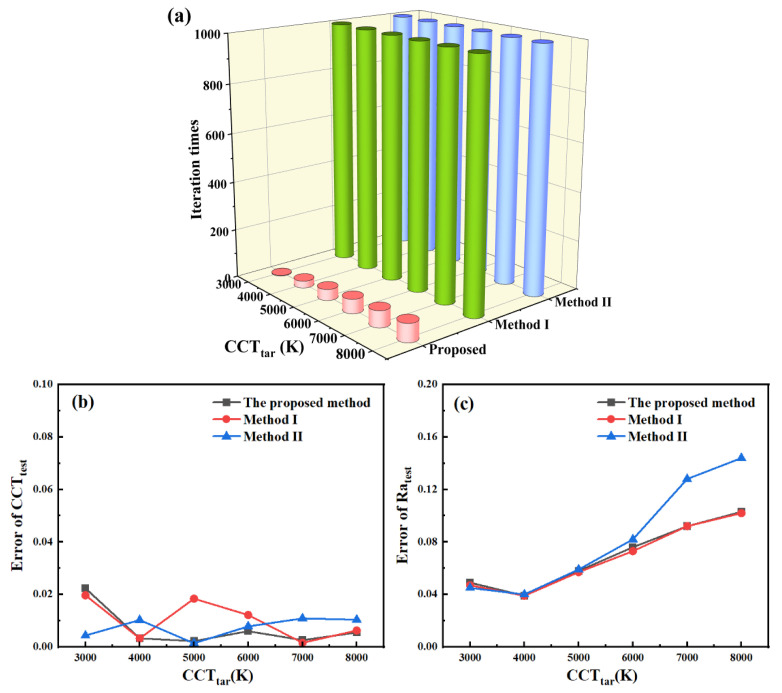
(**a**) Comparison between iteration times of the proposed method, method I, and method II, respectively, under different values of CCT_tar_; (**b**,**c**) comparison between errors of CCT_test_ and Ra_test_ for the proposed method, method I, and method II, respectively, under different values of CCT_tar_.

**Table 1 micromachines-13-00276-t001:** Calculation parameters of spectral optimization using the proposed method.

CCT_tar_ (K)	CCT_test_ (K)	Ra	$\;\delta_{\mathrm{Opt}1}$	$\;\delta_{\mathrm{Opt}2}$	$A_{B}$	$A_{Y}$	$A_{R}$
3000	2924	95.1	200	200	0.11	0.30	0.47
4000	4013	96.1	400	300	0.33	0.30	0.30
5000	5011	94.2	1000	300	0.48	0.30	0.23
6000	6036	92.4	1600	300	0.60	0.30	0.20
7000	7018	90.8	2200	300	0.69	0.30	0.18
8000	8044	89.7	3000	200	0.75	0.30	0.15

**Table 2 micromachines-13-00276-t002:** Calculation parameters of spectral optimization using method I.

CCT_tar_ (K)	CCT_test_ (K)	Ra	$A_{B}$	$A_{Y}$	$A_{R}$
3000	3059	95.3	0.12	0.30	0.46
4000	4013	96.1	0.33	0.30	0.30
5000	4908	94.3	0.46	0.30	0.23
6000	5927	92.7	0.57	0.30	0.18
7000	7011	90.8	0.69	0.30	0.17
8000	7950	89.8	0.72	0.30	0.13

**Table 3 micromachines-13-00276-t003:** Calculation parameters of spectral optimization using method II.

CCT_tar_ (K)	CCT_test_ (K)	Ra	$A_{B}$	$A_{Y}$	$A_{R}$
3000	3013	95.5	0.13	0.30	0.47
4000	4041	96.0	0.34	0.30	0.30
5000	5007	94.1	0.49	0.30	0.24
6000	6047	91.8	0.62	0.30	0.21
7000	6924	87.2	0.58	0.30	0.07
8000	7917	85.6	0.65	0.30	0.05

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
