# Peer review of "An Investigation on CCT and Ra Optimization for Trichromatic White LEDs Using a Dual-Weight-Coefficient-Based Algorithm"

_micromachines, 2022, doi:10.3390/mi13020276_

Round 1

Reviewer 1 Report

The entire results and discussion are very complicated written, and it is hard to understand. I suggest the writing will check by a native English speaker.

  • My major concerns are results and discussion
  • Line 187: it is not clear and so the meaning of “Ratest”
  • Line 212-215: first of all, what is the term “mildly”, second, this phrase is not clear” With the increase of CCTtar, the highest value of Ra can be achieved is decreasing.”
  • Lin3 216: any reference that support this statement?
  • Line 222: “In step 3, different from the optimization of Ratest, there is no 222 need to find the topmost point for CCTtest”. It needs clarification.
  • References need to be updated and more recent works should be added
  • What is the RGB LEDs? It would be better to define the abbreviation before using it.
  • English grammar needs to be checked
  • Results and discussions need to be improved. In its current form, it looks like a report.

Author Response

We thank the reviewers for their useful comments, which have helped us to improve the quality of the paper. Response in detail please refer to the upload file.

Reviewer 2 Report

This work focused on the optimization of CCT (correlated color temperature) and Ra (general color rendering index) of white light generated by combining tri-chromatics (blue LED, YAG: Ce3+ yellow phosphors, and CdSe/ZnSe red quantum dots) and achieved the warm light with CCT of 4000 K and the highest Ra of 96.1. In general, this is an interesting work in this field of research. The manuscript is well written. I recommend its publication in Micromachines after improving the following details.

  1. The authors are suggested to discuss in the manuscript what the general CRI is different than CRI, and what is the benefit of considering general CRI in the light source.
  2. The authors are suggested to provide the quantum yield of both yellow and red phosphors.
  3. The authors have provided the FWHM of blue and red centered spectra, such as 34 nm and 56 nm, respectively, however, just by looking at the line width of these spectra, it looks almost similar. The authors are suggested to correct if there is a typo or mistake in choosing the spectra.
  4. The authors are suggested increasing the font size of Figure 2 so that it would be comfortable for the readers. And please make it a Schematic or Diagram rather than a Figure.
  5. In the explanation of Figure 5, the authors have mentioned that Ratest increases by decreasing the intensity of blue and red emissions. The authors are suggested to provide the logical reasons behind this phenomenon rather than providing the results.
  6. The authors are suggested to follow some related and interesting works: OSA Continuum 2, 1880-1888 (2019); Nanoscale Adv., 2019,1, 1791-1798.
  7. The authors are suggested to eliminate the typos throughout the manuscript. For example, please replace phrases like “synthesize white light” with “fabricate white light” and so on.

Author Response

We thank the reviewers for their useful comments, which have helped us to improve the quality of the paper. Response in detail please refer to the upload file

Round 2

Reviewer 1 Report

The authors addressed all concerns. 

Reviewer 2 Report

This version of manuscript is acceptable for the publication in Micromachines.